# SOCS3 Regulates Dectin-2-Induced Inflammation in PBMCs of Diabetic Patients

**DOI:** 10.3390/cells11172670

**Published:** 2022-08-28

**Authors:** Mohammed J. A. Haider, Zahraa Albaqsumi, Fahd Al-Mulla, Rasheed Ahmad, Fatema Al-Rashed

**Affiliations:** 1Department of Biological Sciences, Faculty of Science, Kuwait University, P.O. Box 5969, Kuwait City 13060, Kuwait; 2Immunology and Microbiology Department, Dasman Diabetes Institute, Al-Soor Street, P.O. Box 1180, Kuwait City 15462, Kuwait; 3Genetics & Bioinformatics, Dasman Diabetes Institute, Al-Soor Street, P.O. Box 1180, Kuwait City 15462, Kuwait

**Keywords:** T2D, Dectin-2, *Candida albicans*, SOCS3, inflammation, PBMC

## Abstract

The C-type lectin receptors (CLRs) Dectin-1 and Dectin-2 are involved in several innate immune responses and are expressed mainly in dendritic cells, monocytes, and macrophages. Dectin-1 activation exacerbates obesity, inflammation, and insulin resistance/type 2 diabetes (T2D). However, the role of Dectin-2 is not clear in T2D. This study aims to evaluate the expression and function of Dectin-2 in peripheral blood mononuclear cells (PBMCs) isolated from diabetic patients and non-diabetic controls. Flow-cytometry and qRT-PCR were performed to evaluate the expression of Dectin-2 in different leukocyte subpopulations isolated from T2D patients (*n* = 10) and matched non-diabetic controls (*n* = 11). The functional activity of Dectin-2 was identified in PBMCs. CRP, IL-1β, and TNF-α concentrations were determined by ELISA. siRNA transfection and Western blotting were performed to assess p-Syk and p-NF-kB expression. siRNA transfection was performed to knock down the gene of interest. Our results show that Dectin-2 expression was the highest in monocytes compared with other leukocyte subpopulations. The expression of Dectin-2 was significantly increased in the monocytes of T2D patients compared with non-diabetic controls. Dectin-2 expression positively correlated with markers of glucose homeostasis, including HOMA-IR and HbA1c. The expression of inflammatory markers was elevated in the PBMCs of T2D patients. Interestingly, SOCS3, a negative regulator of inflammation, was expressed significantly lowlier in the PBMCs of T2D patients. Moreover, SOCS3 expression was negatively correlated with Dectin-2 expression level. The further analysis of inflammatory signaling pathways showed a persistent activation of the Dectin-2-Syk-NFkB pathway that was instigated by the diminished expression of SOCS3. Dectin-2 activation failed to induce SOCS3 expression and suppress subsequent inflammatory responses in the PBMCs of diabetic patients. siRNA-mediated knockdown of SOCS3 in PBMCs displayed a similar inflammatory phenotype to diabetic PBMCs when exposed to Dectin-2 ligands. Altogether, our findings suggest that elevated Dectin-2 and its relationship with SOCS3 could be involved in the abnormal immune response observed in T2D patients.

## 1. Introduction

The prevalence of type 2 diabetes (T2D) is on the rise worldwide. T2D is characterized by metabolic disorders associated with the disruption of glucose homeostasis [1]. Several risk factors are proposed to influence the onset of T2D. However, obesity-mediated chronic and low-grade inflammation remains a major risk factor for insulin resistance and T2D development. The mechanisms related to metabolic inflammation and insulin resistance are poorly understood. Increased production of cytokines/chemokines by inflamed adipose tissue in obesity settings leads to the recruitment of monocytes, neutrophils, and lymphocytes. These immune cells aggravate and sustain the production of inflammatory factors in obese adipose tissue, resulting in insulin resistance [2].

Dectin-1 and Dectin-2 are C-type lectin receptors (CLRs) that recognize β-glucans and mannosylated moieties, respectively, in fungal cell walls and are involved in several innate immune responses against fungal pathogens. During *C. albicans* infection, activation of Dectin-1 and Dectin-2 occurs in circulating monocytes and macrophages. This results in the downstream activation of Src and Syk kinases [3], leading to transcription of nuclear factor-kappa B (NF-κB) and subsequent secretion of pro-inflammatory cytokines [4,5]. Dectin-1 and Dectin-2 are expressed mainly in dendritic cells, monocytes, and macrophages [6,7]. Dectin-1 activation exacerbates the production of inflammatory cytokines, which leads to insulin resistance/T2D [8]. Irregular expression and function of Dectin-1 receptor in T2D patients with poor glycemic control [9]. However, Dectin-2 role in T2D is not clear. In this study, we investigated the expression and function of Dectin-2 in peripheral blood mononuclear cells (PBMCs) isolated from diabetic patients and matched healthy controls.

## 2. Materials and Methods

### 2.1. Patients and Volunteers

A total of 10 T2D patients and 11 matched controls were recruited in this study through the outpatient clinic at Dasman Diabetes Institute, Kuwait. T2D was defined as a fasting plasma glucose level of ≥7.0 mmol/L and use of anti-diabetic drugs. All participants were considered obese with a Body Mass Index (BMI) > 30kg/m2. Participants with serious morbidities, including pulmonary, renal, hepatic, cardiovascular, hematologic or immune diseases, T1D, pregnancy, or malignancy, were excluded.

Clinical characteristics of the study patients are summarized below in Table 1, and bloodwork markers are summarized in Table 2. All participants were informed about the purpose of the study prior to their participation. Written informed consent was obtained from all study participants in accordance with the ethical guidelines of the Declaration of Helsinki and study approval by the Kuwait Ministry of Health Ethical Board (Approval ID#: 1806/2021) and Ethical Review Committee (ERC) of Dasman Diabetes Institute, Kuwait (Approval ID#: RA MoH-2022-002; RA 2010-003). There were no significant differences between non-diabetic and diabetic groups regarding age, body weight, BMI, or fat weight. As anticipated and compared with non-diabetics, diabetic patients showed higher insulin resistance as indicated by their homeostatic model assessment of insulin resistance (HOMA-IR) (*p* < 0.0001) calculated from fasting glucose and insulin secretion. Diabetic patients also had significantly high levels of HbA1c% (*p* = 0.0015), a common marker of patients with uncontrolled T2D.

### 2.2. PBMC Collection

Peripheral blood mononuclear cells (PBMCs) were isolated from non-diabetic and diabetic volunteers in EDTA vacutainer tubes. PBMCs were isolated using the Histo-Paque density gradient method and Sepmate Tube (StemCell Technologies, Vancouver, CN, Canada) [10]. To generate monocytic cells, PBMCs were plated in either 6-well or 24-well plates (Costar, Corning Incorporated, Corning, NY, USA) at 3 × 106 cells/well and cultured in starvation media of RPMI-1640 culture medium (Gibco, Life Technologies, Grand Island, NY, USA) supplemented with 2 mM glutamine (Gibco), 1 mM sodium pyruvate, 10 mM HEPES, 100 ug/mL Normocin, 50 U/ml penicillin, and 50 μg/mL streptomycin (P/S; (Gibco). Cells were incubated at 37 °C (with humidity) in 5% CO_2_ for 3 h. Non-adhered cells were removed, and monocytes adhered to the plate were washed with serum-free culture medium and later incubated for 24 h in RPMI with 2% fetal bovine serum.

### 2.3. Cell Stimulation

For stimulation purposes, medium was replaced with fresh, serum-free medium containing heat-killed *Candida albicans* (HKCA) (Invivogen, Toulouse, France) at MOI = 10 or Furfurman (Invivogen, San Diego, CA, USA) at 20 µg/mL.

### 2.4. Small-Interfering RNA (siRNA) Transfection

Isolated PBMCs were cultured as mentioned previously. Plated cells were transfected with siRNAs using linear polyethylenimines (PEIs) Reagent (Sigma-Aldrich, St. Louis, MO, USA) in accordance with the manufacturer’s instructions [11] with either SOCS3-siRNA (ID: s17189, Invitrogen) or scrambled (control) siRNA (cat. no. 4390844, Invitrogen). Gene knockdown efficiency was examined by qRT-PCR.

### 2.5. Flow Cytometry—Staining of Cell-Surface Markers

To determine the expression of different leukocyte subsets in whole blood, multicolor fluorescence-activated cell sorting (FACS) analysis was conducted using freshly collected whole-blood samples. Briefly, 1 mL lysing buffer was added to 0.1 mL blood samples to eliminate erythrocytes from the remaining peripheral blood leukocyte populations. Cells were washed twice with 1 mL PBS and then incubated with fluorochrome-conjugated mouse anti-human monoclonal antibodies against CD45, CD11b, CD3, CD14, Dectin-2, and isotype-specific respective control antibodies. Details of all antibodies are described in Table 3. Three gating strategies were used to identify and quantify three distinct leukocyte populations, namely: neutrophils (CD45^+^CD14^−^CD11b^+^), monocytes (CD45^+^CD14^+^), and lymphocytes (CD45^+^CD3^+^). Dectin-2 expression was evaluated on different leukocytes according to the gating strategy. Data were collected using a BD FACSCanto 10 flow cytometer and analyzed using BD FACSDiva^TM^ Software 8 (BD Biosciences, Franklin Lakes, NJ, USA). Unstained cells were used to set the quadrant of the negative versus positive gates. Data are presented as mean fluorescence intensity (MFI).

### 2.6. Quantitative Real-Time Polymerase Chain Reaction (qRT-PCR)

Total RNA was extracted using an RNeasy Mini Kit (Qiagen, Valencia, CA, USA) as per the manufacturer’s instructions. cDNA was synthesized from 1 μg total RNA using the high-capacity cDNA reverse transcription kit (Applied Biosystems, Foster city, CA, USA). qRT-PCR was performed on a QuantStudio™ 7 Pro Real-Time PCR System (Applied Biosystems) using a TaqMan^®^ Gene Expression Master Mix (Applied Biosystems). Each reaction contained 50 ng cDNA that was amplified with inventoried TaqMan Gene Expression Assay products (CLEC6A: Assay ID: Hs01073915_m1; CLEC7A: Assay ID: Hs01902549_s1; SOCS3: Assay ID: Hs02330328_s1; GAPDH: Assay ID: Hs03929097_g1; ITGAX/CD11c: Assay ID: Hs00174217_m1; TNFα: Assay ID: Hs01113624_g1; CCL7: Assay ID: Hs00171147_m1; interleukin (IL)-1β assay ID: Hs01555410_m1;interleukin (IL)-10: Assay ID: Hs00961622_m1; and interleukin (IL)-12A assay ID: Hs01073447_m1). The threshold cycle (Ct) values were normalized to the housekeeping gene GAPDH, and the amounts of target mRNA relative to control were calculated using the ΔΔCt-method [12]. Relative mRNA expression was expressed as fold expression over the average of control gene expression. The expression level in control treatment was set at 1. Values are presented as mean ± SEM. Results were analyzed statistically; *p* < 0.05 was considered significant.

### 2.7. Sandwich Enzyme-Linked Immunosorbent Assay (ELISA)

Secreted IL-1β and TNFα protein concentrations were quantified in the supernatants of PBMC-derived monocytes using sandwich ELISA in accordance with the manufacturer’s instructions (R&D systems, Minneapolis, MN, USA).

### 2.8. Western Blotting

Isolated PBMCs were harvested and incubated for 30 min with lysis buffer (Tris 62.5 mM, pH 7.5, 1% Triton X-100, and 10% glycerol). The lysates were centrifuged at 14,000× g for 10 min and the supernatants were collected. Protein concentration in lysates was measured by Quickstart Bradford Dye Reagent, 1x Protein Assay kit (Bio-Rad Laboratories, Inc, Hercules, CA, USA). Protein samples (25 µg) were mixed with sample loading buffer, heated for 5 min at 95 °C, and resolved on 12% polyacrylamide gels using SDS-PAGE. Cellular proteins were transferred to Immuno-Blot PVDF membranes (Bio-Rad Laboratories) by electroblotting. The membranes were then blocked with 5% non-fat milk in PBS for 1 h, followed by incubation with primary antibodies against Syk or p-Syk (MW: 72 kDa), NFκB or p-NFκB (MW: 65 kDa), or β-actin (MW: 42 kDa) at a 1:1000 dilution at 4 °C overnight. All primary antibodies were purchased from Cell Signalling (Cell Signalling Technology, Inc., Danvers, MA, USA). The blots were then washed four times with TBS and incubated for 2 h with HRP-conjugated secondary antibody (Promega, Madison, WI, USA). Immunoreactive bands were developed using an Amersham ECL plus Western Blotting Detection System (GE Health Care, Buckinghamshire, UK) and visualized by Molecular Imager^®^ ChemiDoc^TM^ MP Imaging Systems (Bio-Rad Laboratories).

### 2.9. Statistical Analysis

Statistical analysis was performed using GraphPad Prism software (La Jolla, CA, USA). Data are shown as mean ± standard error of the mean (SEM), unless otherwise indicated. Parametric data were analyzed by unpaired Student’s *t*-test when comparing two groups or one-way ANOVA (if more than two groups) followed by Tukey’s post hoc multiple comparisons test. For non-parametric data, the Wilcoxon–Mann–Whitney U test was used to compare between two groups or the exact Kruskal–Wallis test if comparing more than two groups. Correlation between CLEC6A gene expression and glucose homeostasis parameters (HOMA-IR and HbA1C) was evaluated with Spearman’s correlation coefficients test, while the correlation between parametric association was investigated by Pearson correlation coefficient analysis. For all analyses, data from a minimum of three replicates were used for statistical calculation. A *p* value < 0.05 was considered statistically significant. ns = non-significant, * *p* < 0.05, ** *p* < 0.01, *** *p* < 0.001, and **** *p* < 0.0001.

## 3. Results

### 3.1. Dectin-2 Expression Is Increased on Monocytes in Diabetic Patients

The distribution of Dectin-1 surface expression on blood leukocyte subpopulations has been previously investigated [9]. However, less is known about the distribution of Dectin-2 expression on leukocyte subpopulations in diabetic patients. Surface expression of Dectin-2 in different subpopulations of PBMCs was determined by flow cytometry. Of the total leukocyte population, neutrophils (CD14^−^CD11b^+^) had the highest expression of Dectin-2 (20.5%), while monocytes (CD14^+^) showed lower surface expression, with only 14.3% of cells expressing the receptor, and no surface expression was observed on lymphocytes (CD3^+^ cells) (Figure 1A,B). Although neutrophils showed the highest surface presentation, the mean fluorescence intensity (MFI) of Dectin-2 was significantly the highest in monocytes compared with other leukocyte subpopulations (Figure 1C). Using a similar strategy, we then questioned whether monocytes isolated from diabetic patients express different levels of Dectin-2 on their surface compared with non-diabetic controls. Interestingly, CD45^+^CD14^+^ monocytes taken from diabetic patients showed significantly higher surface expression of Dectin-2 (Figure 1D,E), with similar findings seen in calculated MFI (Figure 1F).

### 3.2. Dectin-2 Gene Expression in Monocytes Correlates with HOMA-IR and HbA1c Levels in Diabetic Patients

To assess the possible role of Dectin-2 in the pathogenesis of diabetes, we investigated the association between Dectin-2 gene (CLEC6A) expression and insulin resistance signatures. Similar to our flow cytometry data, we found significant elevation of CLEC6A gene expression in diabetic patients (*p* < 0.05) compared with non-diabetic controls (Figure 2A). Spearman’s correlation analysis identified a positive correlation of CLEC6A gene expression with HOMA-IR (r = 0.59, *p* < 0.009), calculated from glucose and insulin secretion (Figure 2B), and HbA1c (r = 0.48, *p* < 0.03) (Figure 2C). However, this association was not seen for the expression of Dectin-1 (CLEC7A), as its expression did not differ significantly between diabetic patients and non-diabetic controls (Figure 2D–F). Taken together, these data suggest that diabetic PBMCs show increased expression of CLEC6A and increased Dectin-2 protein expression, which correlates positively with markers of glucose homeostasis.

### 3.3. Dectin-2 Gene Expression Shows a Negative Association with SOCS3

To further explore the functional status of Dectin-2 in diabetic patients, we assessed the effect of Dectin-2 activation on inflammatory markers known to be elevated under diabetic settings using qRT-PCR analysis. Similar to other published data, the elevation of several inflammatory markers is seen in our diabetic population when compared with non-diabatic controls (Figure 3).

The suppressors of the cytokine signaling (SOCS) family are major negative feedback regulators of inflammation that function through the JAK/STAT pathway [13,14]. Previous reports have identified a role for specific CLRs such as Dectin-1 in inducing SOCS proteins [15,16]. To explore this in the context of Dectin-2, we measured the expression levels of SOCS3 gene in the PBMCs. A Significant 3.1-fold decrease (*p* < 0.001) in SOCS3 gene expression was observed in diabetic patients compared with non-diabetic controls (Figure 4A). As expected, the downregulation of SOCS3 in diabetic patients was also found to correlate with increased expression levels of markers of systemic inflammation, including CRP (r = −0.58, *p* < 0.005) and IL-1β (r = −0.51, *p* < 0.017) (Figure 4B,C, respectively). Remarkably, Pearson correlation coefficient analysis indicated a significant negative association between CLEC6A and SOCS3 (r = −0.44, *p* < 0.046) (Figure 4D). Interestingly, the upregulation of STAT3 gene expression in diabetic patients was found to positively correlate with CLEC6A expression (Figure 4E,F), indicating a possible direct/indirect association between Dectin-2 and the activation of downstream signals through the SOCS3-STAT3 pathway.

### 3.4. Dectin-2 Activation Fails to Induce SOCS3 Expression and Suppress Subsequent Inflammatory Responses in the PBMCs of Diabetic Patients

Having identified that diabetic PBMCs express intrinsically lower levels of SOCS3 and higher markers of inflammation, the influence of synthetic agonist-mediated stimulation of Dectin-2 was investigated using heat-killed C. albicans (HKCA) and Furfurman. PBMCs derived from diabetic and non-diabetic patients were stimulated with HKCA or Furfurman and then compared for their gene expression profiles with those of unstimulated PBMCs (vehicle) for SOCS3. In non-diabetic PBMCs, HKCA- and Furfurman-mediated stimulation increased CLEC6A gene expression (approximately 2.0- and 1.4-fold, respectively); however, this was much more pronounced and statistically significant in diabetic PBMCs, where HKCA-mediated and Furfurman-mediated stimulation correlated with an approximately 2.3-fold (*p* < 0.001) and 1.9-fold (*p* < 0.01) increase in CLEC6A gene expression, respectively, compared with unstimulated PBMCs (Figure 5A). HKCA-mediated and Furfurman-mediated stimulation of non-diabetic PBMCs significantly increased the gene expression of SOCS3 (~4.2-fold, *p* < 0.0001 and 2.4-fold, *p* < 0.05, respectively, compared with vehicle controls). However, such an increase in SOCS3 gene expression was not seen in the PBMCs of diabetic patients when stimulated with HKCA and Furfurman (Figure 5B).

The lack of negative inflammatory control was observed in the secretion of inflammatory markers. Supernatant collected from treated samples was processed by ELISA for the secretion of IL-1β and TNFα. The stimulation of PBMCs of diabetic patients with either HKCA or Furfurman significantly increased the secretion of IL-1β (Figure 5C) compared with non-diabetic PBMCs; the magnitude of this difference was much more pronounced for the secretion of TNFα, with the greatest observed difference reported for the HKCA-mediated stimulation of diabetic PBMCs (*p* < 0.0001) (Figure 5D). The observation that IL-1β and TNFα secretion levels were elevated upon HKCA- or Furfurman-mediated stimulation of diabetic PBMCs prompted us to explore whether this was linked to upstream signaling. It is known that the receptor-associated tyrosine kinase Syk is fundamental to CLR signaling, and that one of its downstream targets is NF-κB, a transcription factor crucial to the production of proinflammatory cytokines [17,18]. Therefore, we stimulated PBMCs as described above and then processed total protein by Western blotting for expression of p-Syk and p-NFκB. Interestingly, diabetic PBMCs expressed significantly higher activation of p-Syk at baseline, even in the absence of agonistic stimulation when compared with unstimulated, non-diabetic PBMCs (Figure 5E). Within the diabetic PBMCs group, there was also no significant difference in the expression of p-Syk between unstimulated, HKCA-stimulated, or Furfurman-stimulated PBMCs. Likewise, our data showed sustained p-NF-κB expression in diabetic PBMCs regardless of stimulation compared with non-diabetic PBMCs (Figure 5F).

Together, these data suggest that under Dectin-2 stimulation in diabetic patients, an uncontrolled inflammatory response ensues; this is most likely driven by the lack of SOCS3-mediated modulation of the involved signaling pathway.

### 3.5. SOCS3-Deficient PBMCs Display Similar Inflammatory Phenotype of Diabetic PBMCs When Exposed to Dectin-2 Ligand

To further confirm the role of SOCS3 in Dectin-2-induced inflammatory alterations observed in diabetic patients, primary human monocytes isolated from healthy donors were transfected with SOCS3-specific siRNA, reducing SOCS3 mRNA expression levels by 50–60% (*p* < 0.05) compared with scrambled (control) siRNA (sc-siRNA) **(**Figure 6A). Interestingly, under the influence of Dectin-2 stimulation by HKCA or its synthetic agonist Furfurman, the silencing of SOCS3 in monocytes of healthy, non-diabetic individuals presented similar trends in the production of TNF-α (Figure 6B,C) and IL-1β (Figure 6D), as seen in diabetic PBMCs.

Taken together, these data suggest that the loss of SOCS3 in non-diabetic PBMCs mimics the phenotype observed for diabetic PBMCs when exposed to Dectin-2 ligands.

## 4. Discussion

Dysregulations in CLRs, including Dectin-1, have been associated with the development of inflammatory disorders, such as diabetes mellitus [19,20,21]. Cortez-Espinosa et al. reported that abnormal expression and function of the Dectin-1 receptor in T2D patients with poor glycemic control [9]. However, the possible role of Dectin-2 in the development of T2D has never been evaluated. In this study, it was shown that circulating neutrophils and monocytes both express Dectin-2, the former showing the strongest expression across all leukocyte subpopulations. This observation is in agreement with previous reports on other CLRs [22,23], indicating an important role played by this family of receptors in the innate immune response.

Whereas Dectin-1 expression has been reported on the surface of T-cells and B-cells [22], this was not the case for Dectin-2 in that flow cytometric analysis failed to show Dectin-2 surface expression in CD45^+^CD3^+^ cells. In an earlier study [9], it was demonstrated that the expression of Dectin-1 diminishes under poor glycemic control in diabetic monocytes. The authors hypothesize that this observation could be due to the deregulation of glucose or energy metabolism that seems to decrease Dectin-1 expression. In the present cohort, Dectin-1 gene expression did not differ significantly between diabetic patients and non-diabetic controls. However, Dectin-1 expression was elevated in the adipose tissue of the obese individuals, and it is correlated with metabolic markers [8]. It has been reported that Dectin-1 is expressed in PBMCs from type 2 diabetic patients with poor glycemic control [9]. The possible reasons for the contradiction of our results for Dectin-1 with the previously published reports are because we determined gene expression, not protein, in PBMCs in a small cohort of patients and not in adipose tissue. Of note, our key focus was on Dectin-2 in this study. Dectin-2 expression was significantly elevated in diabetic monocytes compared with non-diabetic controls. Furthermore, this elevation in Dectin-2 expression also correlated with markers of glucose homeostasis (i.e., HOMA-IR and HbA1c levels).

Over the years, the role of SOCS proteins (mainly SOCS1 and SOCS3) as negative regulators of inflammation has been recognized as crucial to the development of insulin resistance and T2D [24]. To this end, we aimed to investigate their role in our cohort in relation to Dectin-2 expression. SOCS3 gene expression was significantly diminished in diabetic patients. The fact that our data show higher inflammatory profiles in our diabatic patients compared with non-diabetic controls and that this upregulation in inflammation is inversely correlated with the expression of SOCS3 falls in line with the view that diabetes entails a state of subacute, chronic, and low-grade inflammation. Specifically, this stems from the absence of inflammation control. The association between the loss of SOCS3 and metabolic syndromes, including T2D, has been addressed previously by several groups [25,26,27]. In a previous study [28], the effect of SOCS3 on insulin resistance in mouse livers was investigated. The authors generated hepatocyte-specific SOCS3-deficient (L-SOCS3 cKO) mice, which unexpectedly exhibited obesity and systemic insulin resistance with age. To this end, they concluded that the absence of SOCS3 expression in the liver promotes systemic insulin resistance by mimicking chronic inflammation. The loss of SOCS3 expression is seen not only in diabetic settings but also in other metabolic syndromes, such as nonalcoholic fatty liver disease (NAFLD). Moreover, the loss of SOCS3 is believed to be driven by the alteration of SOCS3 methylation states triggered by specific lifestyles, including obesity [29,30]. Even though this is a very interesting perspective to further investigate, it became clear to us that there is another dynamic we need to pay attention to. During our work, further correlation analysis pointed out that this reduction in gene expression of SOCS3 was also inversely associated with the expression of CLEC6A, the gene encoding Dectin-2. We also found that CLEC6A expression correlated positively with that of STAT3, indicating overactivation in inflammatory signaling due to the loss of immunomodulation by SOCS3.

Candidiasis is amongst the most typically reported infections in diabetic individuals [31,32,33]. Although *C. albicans* is considered a commensal micro-organism, in diabetic settings, it readily becomes an opportunistic pathogen [31]. Taking into consideration our previous observation regarding the loss of SOCS3 and the activation of Dectin-2 and downstream STAT signaling, we questioned how PBMCs isolated from T2D patients would differ from non-diabetic PBMCs in their response to Dectin-2 stimulation. To perform this, we challenged isolated PBMCs with either HKCA or a Dectin-2-specific synthetic agonist, Furfurman. Dectin-2 stimulation did not change the inflammatory outcome in non-diabetic controls; however, the opposite was observed for diabetic patients. Using Western blot analysis, we identified a constant upregulation in the phosphorylation of Syk, which subsequently induced a constant upregulation in the phosphorylation of the transcription factor NF-kB. The regulation and control of adaptive immunity against fungal pathogens such as *C. albicans* is controlled at higher levels compared with other pathogens; this could be due to the nature of the coexistence and the similar cellular structure mammalian cells share with fungi. An earlier study showed that Syk signaling triggered by Dectin-1 induces two independent signaling pathways, one through Syk and the other through Raf-1 kinase, which converge at the level of NF-kB activation to control Th1- and Th17-polarising cytokines in response to *C. albicans* [34]. This level of control is reflected poorly in PBMCs isolated from diabetic patients. Not only did we observe elevated expression at basal levels, but we also observed no response to stimulation, hence indicating a loss of proper innate responses to fungal pathogens. We further sought to investigate if the loss of SOCS3 in non-diabetic controls shows similar inflammatory dysregulation. Indeed, silencing SOCS3 gene expression in mononuclear monocytes isolated from non-diabetic individuals mimicked the lack of control seen in T2D patients upon Dectin-2 stimulation, thereby indicating a lack of control towards opportunistic pathogens. However, more work is needed to better understand the complexity of the signaling crosstalk between SOCS3 and Dectin-2. Even though this study is the first of its kind to address the dynamic relation between Dectin-2 receptor and SOCS3 expression in T2D patients, there is a key limitation to this study. The study included a small population and, therefore, caution may be warranted while interpreting the results for generalizability.

## 5. Conclusions

In conclusion, our results suggest that in diabetic settings, Dectin-2 expression positively correlated with insulin resistance signatures. T2D patients’ PBMCs show increased expression of inflammatory markers and decreased expression of SOCS3, which is negatively correlated with Dectin-2. Dectin-2 activation fails to induce SOCS3 expression and suppress subsequent inflammatory responses in diabetic PBMCs. SOCS3–siRNA-treated PBMCs show a similar inflammatory phenotype to that of diabetic PBMCs when exposed to Dectin-2 ligands. Collectively, these findings suggest that a Dectin-2/SOCS3 relation could be partly involved in the abnormal immune responses observed in T2D patients, and therein may be implicating a potential target for therapeutic intervention.

## Figures and Tables

**Figure 1 cells-11-02670-f001:**
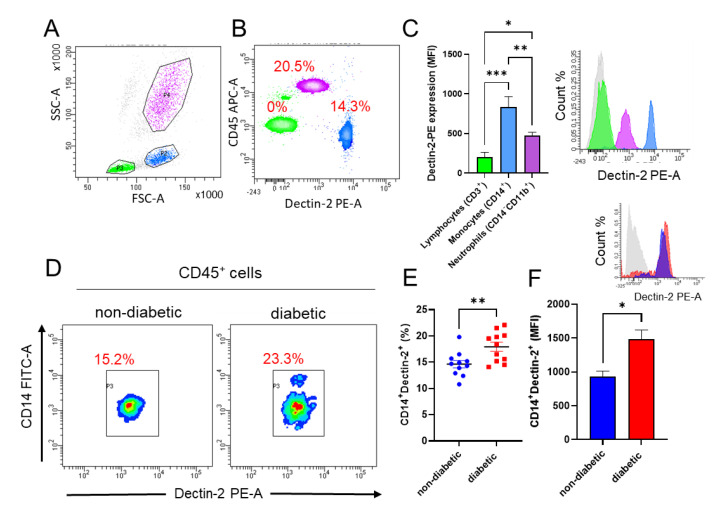
Increased Dectin-2 expression in diabetic patients’ PBMCs. Flow cytometric analysis was conducted to identify the expression levels of Dectin-2 on different leukocyte subsets in PBMCs isolated from non-diabetic and diabetic patients. (**A**) Representative dot plot identifying three leukocyte subsets. (**B**) Representative dot plot of Dectin-2 expression in different leukocyte subsets. (**C**) Bar graph analysis of Dectin-2 expression calculated as MFI with representative histogram. (**D**) Representative density plot analysis of CD45^+^CD14^+^Dectin-2^+^ population in PBMCs isolated from non-diabetic and diabetic patients. (**E**) The percentage of CD45^+^CD14^+^Dectin-2^+^ population in PBMCs isolated from non-diabetic and diabetic patients. (**F**) Bar graph analysis of CD45^+^CD14^+^Dectin-2^+^ expression calculated as MFI with representative histogram. Gray (non-stained), green (lymphocytes), blue (monocytes), and purple (neutrophils). Data are presented as mean ± SEM with a minimum of n = 3. Statistical significance was assessed by Student’s *t*-test. * *p* < 0.05, ** *p* < 0.01, and *** *p* < 0.001.

**Figure 2 cells-11-02670-f002:**
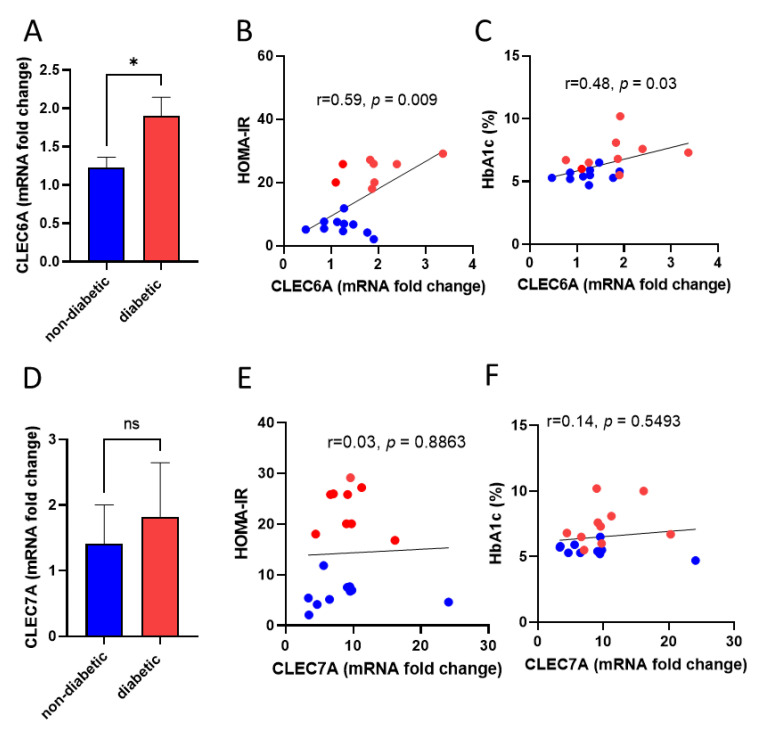
Dectin-2 expression in monocytes correlates with HOMA-IR and HbA1c levels in diabetic patients. (**A**) Dectin-2 (CLEC6A) gene expression was determined by qRT-PCR in PBMCs of non-diabetic and diabetic patients. (**B**,**C**) Spearman’s correlation analysis was performed to determine correlation between CLEC6A and HOMA-IR andHbA1c. (**D**) Dectin-1 (CLEC7A) gene expression was determined d by qRT-PCR. (**E**,**F**) Spearman’s correlation analysis was also performed to determine correlation between CLEC7A and HOMA-IR and HbA1c. Data are presented as mean ± SEM with a minimum of n = 3. Statistical significance was assessed by Student’s *t*-test or Spearman’s correlation test. * *p* < 0.05 and ns = non-significant.

**Figure 3 cells-11-02670-f003:**
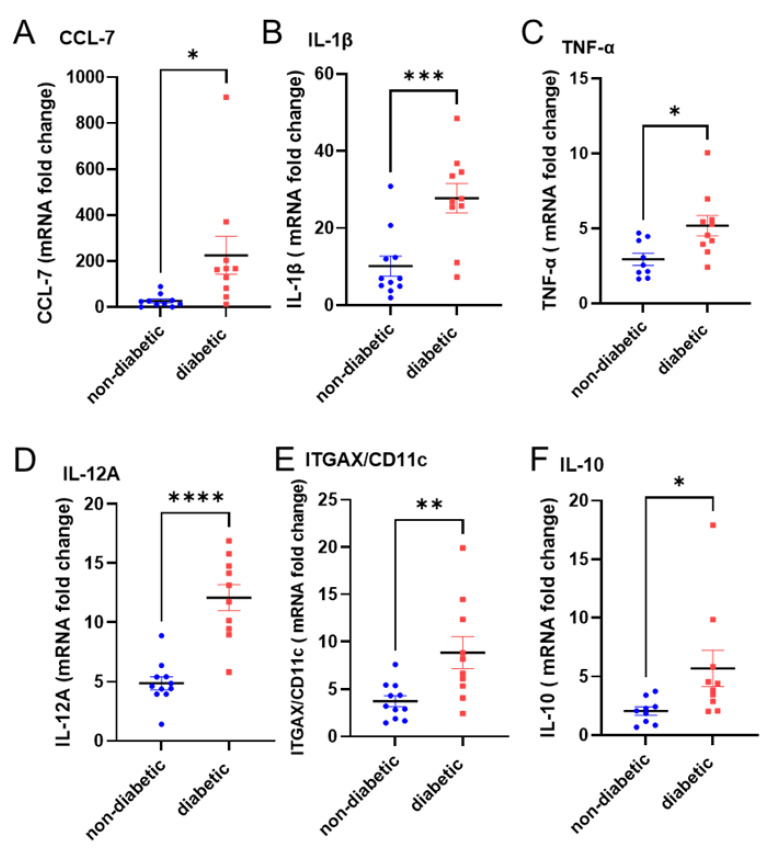
PBMCs from diabetic patients show elevated gene expression of markers of inflammation. PBMCs were isolated from non-diabetic and diabetic patients and processed for RNA extraction. Quantification of gene expression was performed by qRT-PCR. Results are shown for (**A**) CCL-7, (**B**) IL-1β, (**C**) TNF-α, (**D**) IL-12A, (**E**) ITGAX/CD11c, and (**F**) IL-10 and are expressed as the mean ± SEM mRNA fold change compared with housekeeping control (GAPDH). Statistical significance was assessed by Student’s *t*-test. * *p* < 0.05, ** *p* < 0.01, *** *p* < 0.001, **** *p* < 0.0001.

**Figure 4 cells-11-02670-f004:**
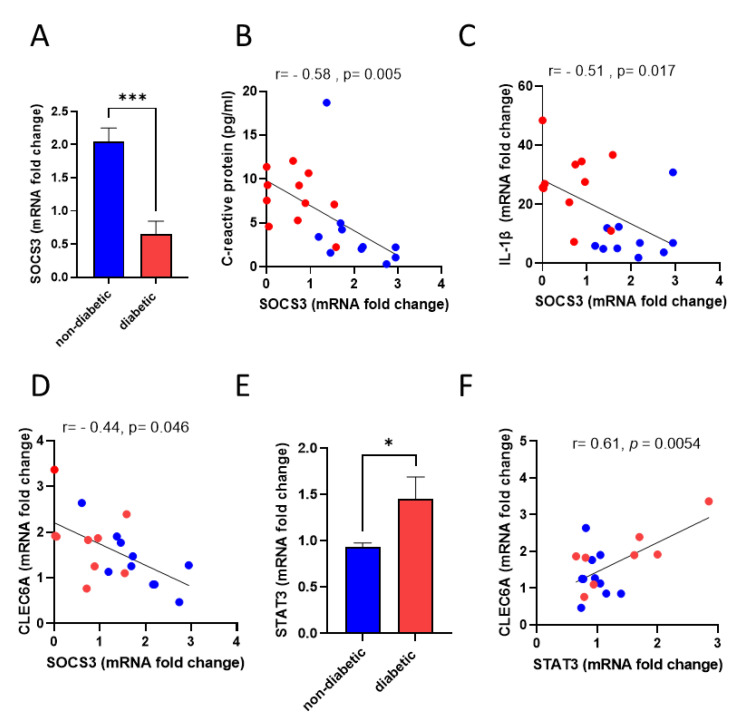
SOCS3 gene expression is decreased in diabetic patients and shows association with inflammatory markers. PBMCs were isolated from non-diabetic and diabetic patients and processed for RNA extraction. (**A**) SOCS3 gene expression. (**B**,**C**) Correlation analysis for inflammatory markers; CRP and IL-1β withSOCS3. (**D**) Correlation analysis for SOCS3 withCLEC6A gene expression. (**E**) STAT3 gene expression. (**F**) Correlation analysis between STAT3 gene expression and CLEC6A. Data are expressed as mean ± SEM mRNA fold change compared with housekeeping control (GAPDH). Statistical significance was assessed by Student’s *t*-test or Pearson correlation coefficient analysis. * *p* < 0.05 and *** *p* < 0.001.

**Figure 5 cells-11-02670-f005:**
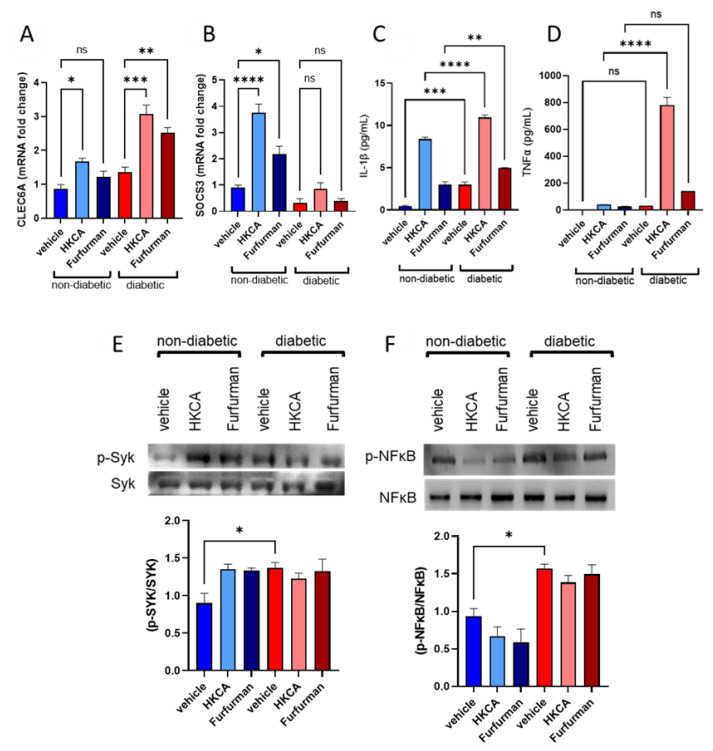
Stimulation of diabetic PBMCs with Dectin-2 synthetic agonists associates with higher expression of CLEC6A and inflammatory markers but lower expression of SOCS3. PBMCs from non-diabetic and diabetic patients were stimulated with either HKCA or Furfurman to investigate the effect of Dectin-2-induced inflammation. (**A**,**B**) qRT-PCR analysis of CLEC6A and SOCS3 gene expression. (**C**,**D**) ELISA results for IL-1β and TNF-α secretion in supernatants. (**E**,**F**, upper panels) Western blot analysis of p-Syk and p-NFκB protein expression corrected to their respective total protein expression with representative immune blots. (**E**,**F**, lower panels) Densitometric analysis of p-Syk and p-NFκB protein expression. Data are presented as the mean ± SEM with minimum n = 3. Statistical significance was assessed by one-way ANOVA followed by Tukey’s post hoc multiple comparisons test where *p* < 0.05 was considered significant. ns = non-significant, * *p* < 0.05, ** *p* < 0.01, *** *p* < 0.001, and **** *p* < 0.0001.

**Figure 6 cells-11-02670-f006:**
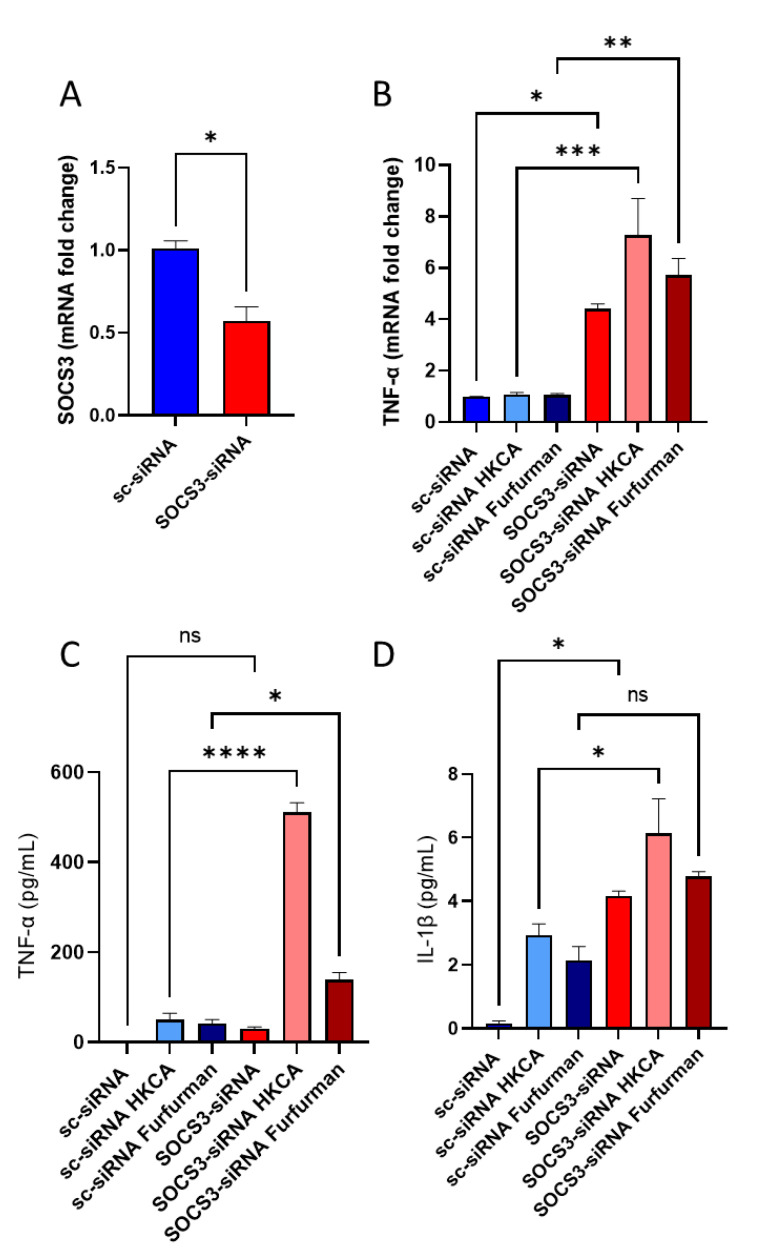
siRNA-mediated silencing of SOCS3 in non-diabetic PBMCs mimics diabetic inflammatory responses. PBMCs from non-diabetic patients were transfected with scrambled-siRNA (negative control) or SOCS3-siRNA. qRT-PCR was preformed to measure (**A**) SOCS3 gene expression. SOCS3-silenced cells were treated with the Dectin-2 stimuli HKCA or Furfurman, and the expression of inflammatory responses was observed by (**B**) TNF-α gene expression. Collected supernatants were processed by ELISA for protein secretion of (**C**) TNF-α and (**D**) IL-1β. Data are expressed as the mean ± SEM with minimum *n* = 3. Statistical significance was assessed by Student’s *t*-test and one-way ANOVA followed by Tukey’s post hoc multiple comparisons test where *p* < 0.05 was considered significant. ns = non-significant, * *p* < 0.05, ** *p* < 0.01, *** *p* < 0.001, and **** *p* < 0.0001.

**Table 1 cells-11-02670-t001:** Patient characteristics.

Characteristic	Non-Diabetic *n* = 11 (6M/5F)	Diabetic *n* = 10 (3M/7F)	*p* Value
Age (years)	44.2 ± 11.5	48 ± 12.5	0.3035
Weight (kg)	86.8 ± 18.6	89.1 ± 12.5	0.7454
Height (cm)	1.6 ± 0.08	1.69 ± 0.06	0.9365
BMI (kg/m^2^)	30.4 ± 6.2	31.1 ± 3.4	0.7455
Waist circumference(inch)	103.5 ± 17.9	102.5 ± 8.12	0.8950
Hip circumference (inch)	107.0 ± 12.4	114.8 ± 10.8	0.1903
Fat weight (kg)	34.4 ± 7.4	36.9 ± 5.0	0.3245

**Table 2 cells-11-02670-t002:** Patient bloodwork.

Blood work	Non-Diabetic *n* = 11 (6M/5F)	Diabetic *n* = 10 (3M/7F)	*p* Value
BP/ systolic (mmHg)	123.5 ± 13.4	132 ± 10.3	0.1630
BP/diastolic (mmHg)	81.1 ± 9.1	82 ± 12.2	0.9745
HR	77 ± 21.4	114.8 ± 14.4	0.6057
Triglycerides (mmol/L)	1.4 ± 0.86	1.5 ± 0.68	0.6333
Total cholesterol (mmol/L)	4.8 ± 0.77	5.0 ± 0.79	0.4856
HDL cholesterol (mmol/L)	1.1 ± 0.14	1.2 ± 0.31	0.5354
Fasting glucose (mmol/L)	5.2 ± 0.48	7.8 ± 1.6	**0.0001**
Insulin Con. (mu/L)	5.1 ± 1.6	59.3 ± 40.2	**0.0004**
HOMA-IR	3.4 ± 3.6	23.1 ± 4.4	**<0.0001**
HbA1c (%)	5.5 ± 0.48	7.4 ± 1.5	**0.0015**
CRP (pg/mL)	3.9 ± 5.6	7.4 ± 2.8	**<0.0001**

**Table 3 cells-11-02670-t003:** Antibodies list.

Antibody	Host/Isotype	Cat #	Manufacturer
Anti-CD3- PE-Cy7	IgG1	563423	BD Pharmingen™
Anti-CD45 VioBlue	IgG2a	130-113-684	Miltenyi Biotec
Anti-CD14- FITC	IgG2a	555397	BD Pharmingen™
Anti-CD11b (D12)-APC	IgG2a	340936	BD Biosciences
Dectin-2/CLEC6A PE	IgG1	FAB3114P	R&D systems

## Data Availability

Not applicable.

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
