# Peer review of "SOCS3 Regulates Dectin-2-Induced Inflammation in PBMCs of Diabetic Patients"

_cells, 2022, doi:10.3390/cells11172670_

Round 1

Reviewer 1 Report

This is an interesting paper on the correlation between Dectin-2/SOCS3 and the abnormal immune responses in PBMCs from T2D patients.

There are some issues should be addressed:

1. All participants were considered obese with a Body Mass Index (BMI) > 30kg/m2. Why did the individuals with a BMI within the normal range are excluded in the study? Could the authors provide the evidences to show that the influence of obesity was excluded.

2. The study mainly utilities the PBMCs from clinical samples to support the results. However, the sample size in the study is small, the conclusion seems to be overstatement. 

Author Response

We thank the reviewer for your thoughtful comments. We have now performed some of the suggested additional experiments and included data in our revised manuscript.

Please see attached point by point responses to comments.

Reviewer 2 Report

This is an interesting and novel investigation showing increased expression of Dectin-2 in type 2 diabetic patients and its correlation with inflammatory markers and lack of functional SOCS3. It is relevant and important to the field of T2DM mechanism and pathogenesis.

There are some minor issues in the manuscript that the authors should consider correcting:

1. In the Introduction (line 48) there is no reference to support the inflammatory response in obesity.

2. In the Introduction and the Discussion the authors attach great importance to the role of Dectin-1 in T2DM and yet lines 57-58 rely on a single reference to back this up. Furthermore on line 361, the references 18,19 are inflammatory disorders but are not T2DM. The authors need to give a fuller explanation of the background that suggests Dectins might be involved in T2DM.

3. Related to the point above is the observation by the authors in Figure 2 that  Dectin-1 shows no change or correlation with inflammatory markers in T2DM patients in this investigation. This is not discussed at all by the authors even though it contradicts their stated justification for the study.

4. Fig 1A identifies 3 subset populations of leukocytes but does not tell the reader what each colour represents or how the populations were defined. This should be in the legend. 

5. Fig1B and text (lines 186-187) claims no surface expression of Dectin-2 and yet Fig 1C appears to show low expression.

6. Lines 216-218 need rephrasing. One cannot "express higher levels of Dectin-2 at both gene and protein levels" - could have increased expression of Dectin-2 gene and increased Dectin-2 protein expression

Author Response

(The authors gave the same response as above.)
